# Adrenomedullin-CALCRL axis controls relapse-initiating drug tolerant acute myeloid leukemia cells

Clément Larrue[1,2], Nathan Guiraud[1], Pierre-Luc Mouchel [1,3], Marine Dubois[1], Thomas Farge[1], Mathilde Gotanègre[1], Claudie Bosc[1], Estelle Saland[1], Marie-Laure Nicolau-Travers[1,3], Marie Sabatier[1], Nizar Serhan[1], Ambrine Sahal[1], Emeline Boet[1], Sarah Mouche[2], Quentin Heydt[1], Nesrine Aroua[1], Lucille Stuani[1], Tony Kaoma[4], Linus Angenendt [5], Jan-Henrik Mikesch[5], Christoph Schliemann[5], François Vergez[1,3], Jérôme Tamburini[2,6,7,8], Christian Récher [1,3,8] & Jean-Emmanuel Sarry [1✉]

Drug tolerant/resistant leukemic stem cell (LSC) subpopulations may explain frequent relapses in acute myeloid leukemia (AML), suggesting that these relapse-initiating cells (RICs) persistent after chemotherapy represent bona fide targets to prevent drug resistance and relapse. We uncover that calcitonin receptor-like receptor (CALCRL) is expressed in RICs, and that the overexpression of CALCRL and/or of its ligand adrenomedullin (ADM), and not CGRP, correlates to adverse outcome in AML. CALCRL knockdown impairs leukemic growth, decreases LSC frequency, and sensitizes to cytarabine in patient-derived xenograft models. Mechanistically, the ADM-CALCRL axis drives cell cycle, DNA repair, and mito-chondrial OxPHOS function of AML blasts dependent on E2F1 and BCL2. Finally, CALCRL depletion reduces LSC frequency of RICs post-chemotherapy in vivo. In summary, our data highlight a critical role of ADM-CALCRL in post-chemotherapy persistence of these cells, and disclose a promising therapeutic target to prevent relapse in AML.

[1] Centre de Recherches en Cancérologie de Toulouse, UMR1037, Inserm, Université de Toulouse 3 Paul Sabatier, Equipe Labellisée LIGUE 2018, F-31037 Toulouse, France. [2] Geneva University Hospitals, University of Geneva, Geneva, Switzerland. [3] Service d'Hématologie, Institut Universitaire du Cancer de Toulouse-Oncopole, CHU de Toulouse, F-31100 Toulouse, France. [4] Proteome and Genome Research Unit, Department of Oncology, Luxembourg Institute of Health, L-1445 Strassen, Luxembourg. [5] Department of Medicine A, University Hospital Münster, Münster, Germany. [6] Institut Cochin, Département Développement, Reproduction et Cancer, CNRS UMR8104, Inserm U1016, Paris, France. [7] Université Paris Descartes, Faculté de Médecine Sorbonne Paris Cité, Paris, France. [8] These authors contributed equally: Jérôme Tamburini, Christian Récher. ✉email: jean-emmanuel.sarry@inserm.fr

Despite improvements in the complete response rate obtained after conventional chemotherapy, the overall survival of acute myeloid leukemia (AML) patients is still poor, due to frequent relapses caused by chemotherapy resistance. Although novel targeted therapies are holding great promises, eradicating drug tolerant/resistant AML cells after chemotherapy remains the major challenge in the treatment of AML.

AML arises from self-renewing leukemic stem cells (LSCs) that can repopulate human AML when assayed and xenografted in immunocompromised mice[1]. Even though these cells account for a minority of leukemic burden, gene signatures associated with a stem cell phenotype or function have an unfavorable prognosis in AML[2–6]. Clinical relevance is evidenced by the enrichment in LSC-related gene signatures in AML specimens at the time of relapse compared to diagnosis[7,8]. While it has been initially shown that LSCs may be less affected by chemotherapy than more mature populations[9–11], recent works demonstrated that the anti-AML chemotherapy cytarabine (AraC) might deplete the LSC pool in patient-derived xenograft (PDX) models[12,13]. These results suggest the coexistence of two distinct LSC populations, one chemosensitive and thus eradicated by conventional treatments and one that is chemoresistant, persists and might induce relapse in AML (relapse-initiating drug-resistant leukemic stem cells, RIC). Thus, a better phenotypical and functional characterization of RICs is crucial for the development of new AML therapies aiming at reducing the risk of relapse.

Although it was first proposed that the LSC-compartment was restricted to the CD34$^+$CD38$^-$ subpopulation of human AML cells[1,10], several studies subsequently demonstrated that LSCs are phenotypically heterogeneous when assayed in NSG mice[2,14–16]. These observations highlight that more functional studies are needed to better and more fully characterize LSCs, particularly under the selection pressure imposed by chemotherapy. Eradicating LSCs without killing normal hematopoietic stem cells (HSCs) depends on the identification of therapeutically relevant markers that are overexpressed in the AML compartment. In recent years, tremendous research efforts led to the identification of several cell surface markers such as CD47, CD123, CD44, TIM-3, CD25, CD32, or CD93 discriminating LSCs from HSCs[17–22]. In addition, it has been proposed that LSCs also have both a specific decrease in their reactive oxygen species (ROS) content and an increase in BCL2-dependent oxidative phosphorylation (OxPHOS), revealing a vulnerability that can be exploited through treatment with BCL2 inhibitors such as venetoclax[23,24]. This is consistent with several studies demonstrating that mitochondrial OxPHOS status contributes to drug resistance in myeloid leukemia[12,25–27]. Taken together, these results suggest that specific characteristics of LSCs can be exploited to develop targeted therapeutic approaches.

Here, we report that the cell surface G protein-coupled receptor (GPCR) family calcitonin receptor-like receptor (CALCRL) and its ligand adrenomedullin (ADM) are expressed in AML cells and that CALCRL sustains LSC function. High expression of both CALCRL and ADM is predictive of an unfavorable prognosis in a cohort of 179 AML patients. Moreover, depletion of CALCRL abrogates leukemic growth and dramatically induces cell death in vivo. Transcriptomic and functional analyses show that CALCRL drives E2F1, BCL2, and OxPHOS pathways involved in the chemoresistance. Furthermore, we observed that cell surface expression of CALCRL is enriched after AraC treatment in PDX models as well as after intensive chemotherapy in AML patients. Limiting dilution analyses coupled to genetic manipulation demonstrates that RICs are critically dependent on CALCRL for their maintenance. Altogether, our findings demonstrate that CALCRL is a new RIC player with a critical effector role in both their stemness and chemoresistance, and thus is a relevant target to eradicate this specific cell population.

## Results

### The receptor CALCRL and its ligand adrenomedullin are expressed in AML cells and associated with a poor outcome in patients.

Using a clinically relevant chemotherapeutic model, we and others previously demonstrated that LSCs are not necessarily enriched in post-AraC residual AML, suggesting LSCs include both chemosensitive and chemoresistant stem cell subpopulations[12,13]. In order to identify new vulnerabilities in the chemoresistant LSC population that might be responsible for relapse, we analyzed transcriptomic data from three different studies that (Fig. 1a and Table S1): (i) identified 134 genes overexpressed in functionally defined LSC compared with a normal HSC counterpart (Eppert et al., 2011; GSE30377)[2]; (ii) uncovered 114 genes of high expression associated with poor prognosis in AML (the Cancer Genome Atlas, AML cohort, 2013); and (iii) selected 536 genes overexpressed at relapse compared to pairwise matched diagnosis samples (Hackl et al., 2015; GSE66525)[7]. Surprisingly, we found one unique gene common to these three independent transcriptomic datasets: CALCRL, encoding a G protein-coupled seven-transmembrane domain receptor poorly documented in cancer that has been recently described as associated with a poor prognosis in AML[28]. Using four independently published cohorts of AML patients (TCGA AML cohort; GSE12417; GSE14468; BeatAML cohort), we observed that patients with high CALCRL expression had a shorter overall survival (Fig. 1b and Fig. S1a) and are more refractory to chemotherapy (Fig. S1b) compared to patients with low CALCRL expression. This correlated with a higher expression in complex versus normal karyotypes (Fig. S1c). Furthermore, CALCRL gene expression was significantly higher at relapse compared to diagnosis in patients treated with intensive chemotherapy (Fig. 1c). CALCRL expression was also higher in the leukemic compartment compared with normal hematopoietic cells, and more specifically in the LSC population as both functionally (Fig. 1d) and phenotypically (Fig. S1d) defined, compared with the AML bulk population. Interestingly, CALCRL expression was higher in the more immature AML subtypes according to FAB stratification, suggesting that CALCRL is a marker of cell immaturity (Fig. S1e). Using flow cytometry, we determined that CALCRL was expressed at the cell surface (Fig. S1f), more markedly in leukemic compared to normal CD34$^+$ hematopoietic progenitor cells (Fig. S1G; see Table S2 for mutational status of patients). Of note, CALCRL expression did not correlate with any most-found mutations (Fig. S1h). Next, we assessed the expression of ADM, a CALCRL ligand already described in several solid cancers[29,30]. The ADM gene is overexpressed in AML cells compared to normal cells (Fig. S1j, k), although its expression is not altered in AML patients at relapse compared to diagnosis (Fig. S1l) and is not linked to mutational status (Fig. S1i). Using a combination of western blotting, confocal microscopy, and RNA microarray, we have established that CALCRL, its three co-receptors RAMP1, RAMP2 and RAMP3, as well as ADM (but not CGRP, another putative CALCRL ligand) are expressed in all the tested AML cell lines and primary AML samples (Fig. S1m–r). Moreover, analysis of three independent cohorts (TCGA, Verhaak et al. and BeatAML) confirmed that only ADM was highly expressed in primary samples compared to CALCA and CALCB (two genes encoding CGRP) that were not expressed or poorly expressed (Fig. S1s). Next, we addressed the impact of CALCRL and ADM protein levels at diagnosis on patient outcome. Using IHC analyses, we observed that increasing protein levels of CALCRL or ADM were associated with decreasing complete remission rates, inferior 5-year overall survival and event-free survival (EFS) in a cohort of 179 intensively treated AML patients (Fig. 1e, f). When patients were clustered into 4 groups according to CALCRL and ADM expression (low/low vs low/high vs high/

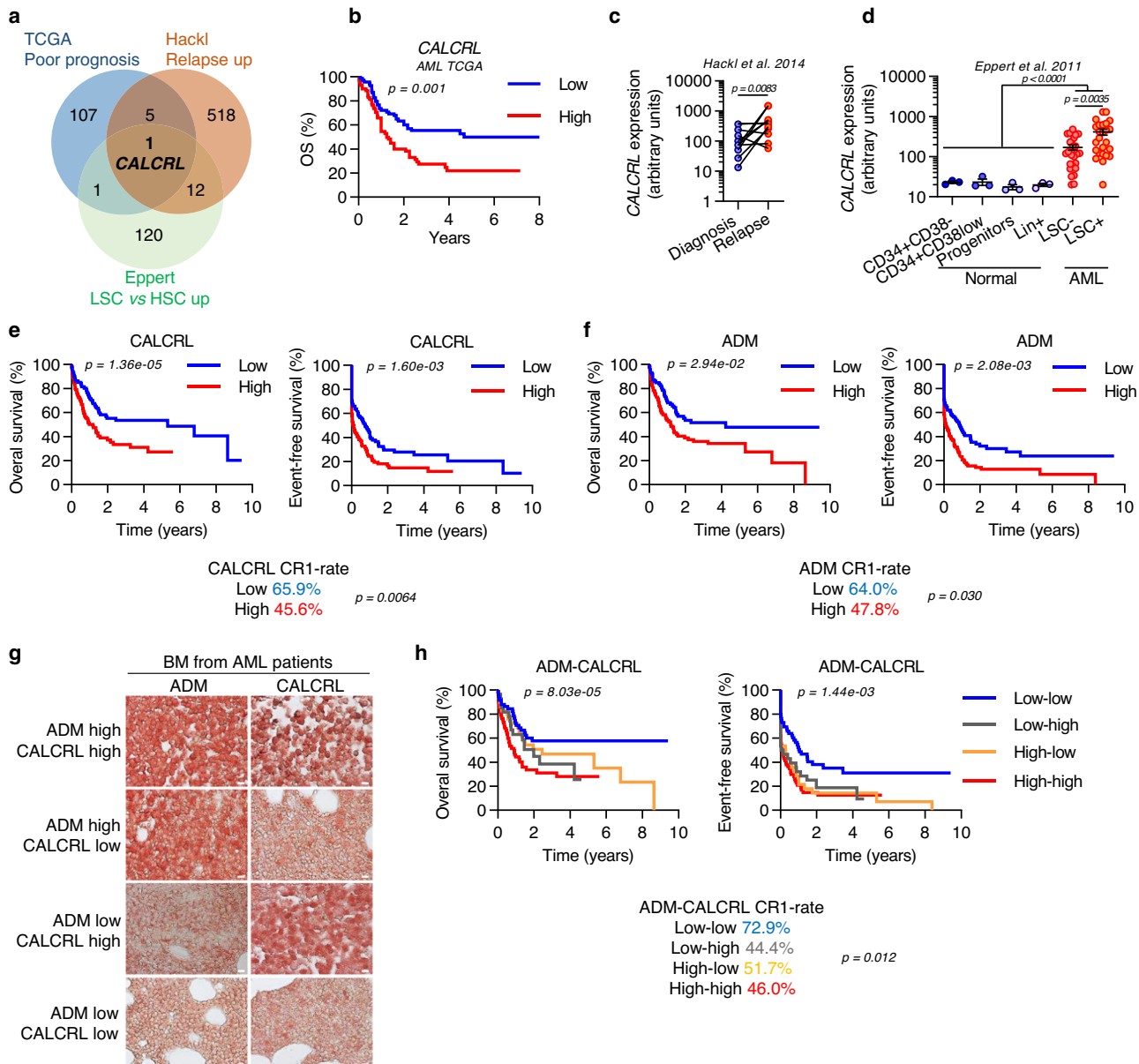

**Fig. 1 Expression of CALCRL and its ligand adrenomedullin and impact on patient outcome in AML. a** Genes overexpressed and associated with poor prognosis, relapse, or LSC vs HSC according to TCGA, Hackl et al.[7] and Eppert et al.[2] studies, respectively. **b** Impact of *CALCRL* gene expression on overall survival in the TCGA cohort (Log-rank (Mantel-Cox) test). Data were dichotomized according to median value for visualization purpose. **c** *CALCRL* expression at diagnosis and relapse after intensive chemotherapy according to Hackl et al.[7] (unpaired *t*-test). **d** *CALCRL* gene expression in normal compartments CD34+CD38−, CD34+CD38low, progenitors, and Lin+ compared with leukemic LSC− and LSC+ compartments according to Eppert et al.[2] (unpaired *t*-test) **e, f** Overall survival (OS) and event-free survival (EFS) according to CALCRL and ADM H-scores. OS and EFS were regressed against CALCRL and/or ADM H-scores using univariate or multivariate Cox model. Data were dichotomized for visualization purpose. **g** Representative IHC micrographs of CALCRL and ADM expression in pretherapeutic BM from AML patients. A picture of each primary sample was taken to make the quantifications. Scale bars correspond to 10 µm. **h** OS and EFS according to CALCRL and ADM H-scores. OS and EFS were regressed against CALCRL and/ or ADM H-scores using univariate or multivariate Cox model. Data were dichotomized for visualization purpose.

low vs high/high; Fig. 1G and Table S3), we observed that the CALCRL^high/ADM^high group was associated with the lowest overall survival and that high expression of only *CALCRL* or *ADM* also correlated to reduced EFS and complete remission rate (Fig. 1h). Next, we addressed the protein level of CGRP using IHC analyses and we detected a slight diffuse signal of this protein, suggesting a paracrine secretion of CGRP in AML. However, we demonstrated that protein levels of CGRP had no impact on 5-year overall survival and EFS (Fig. S2a–c), indicating that ADM was likely the main driver of CALCRL activation in AML.

All these data supported the hypothesis that the ADM-CALCRL axis is activated in an autocrine-dependent manner and associated with a poor prognosis in AML.

**The CALCRL-ADM axis is required for cell growth and survival.** Next, we investigated the impact of deregulated CALCRL-ADM axis in cell proliferation and survival. CALCRL depletion was associated with a decrease in blast cell proliferation (Fig. 2a) and an increase in cell death (Fig. 2b) in three AML (MOLM-14, OCI-AML2, OCI-AML3) cell lines. Furthermore, ADM-targeting

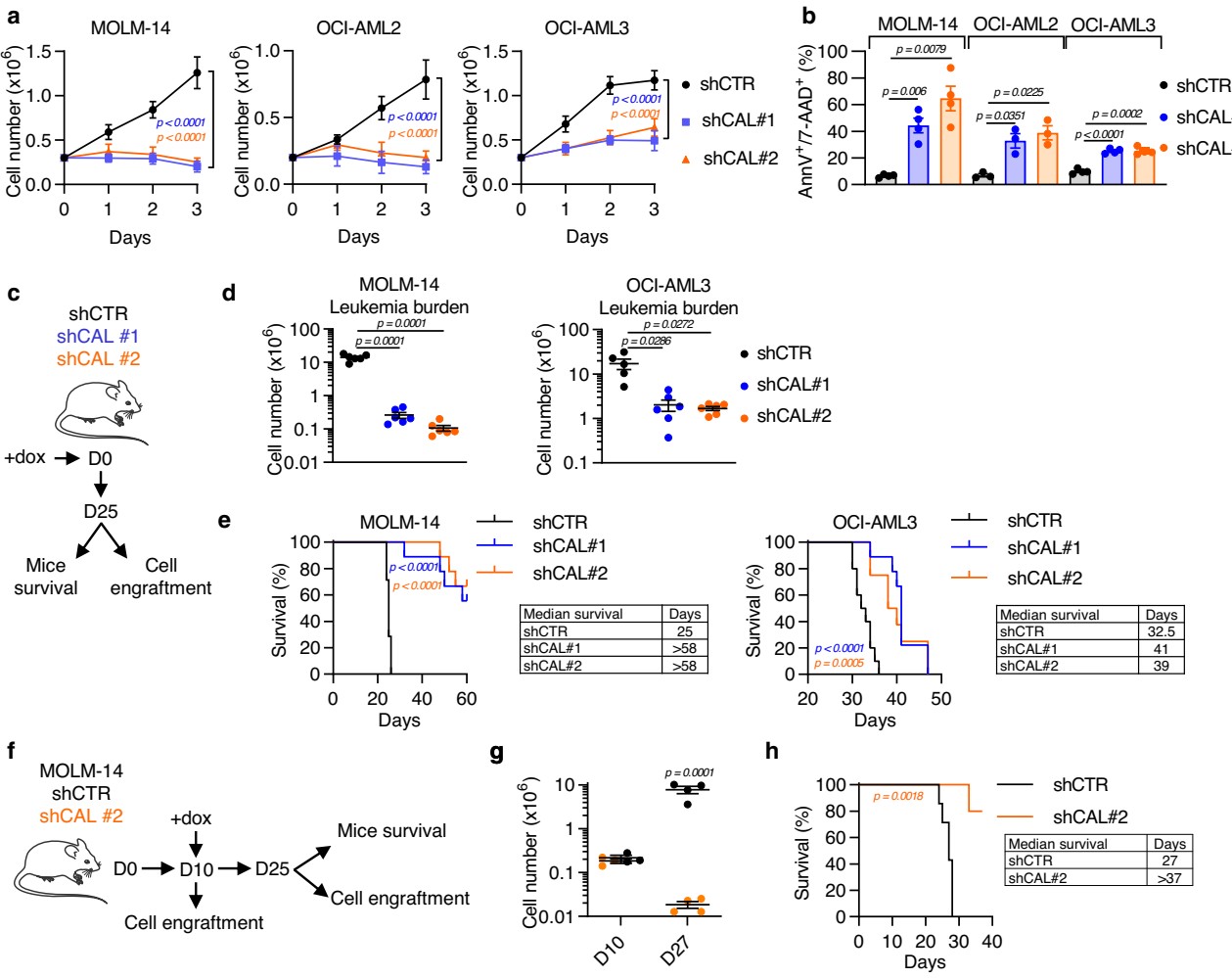

**Fig. 2 CALCRL depletion reduces AML cell growth in vitro and in vivo. a–c** Control shRNA or shCALCRL were expressed in MOLM-14, OCI-AML2, or OCI-AML3 cell lines. (**a**) Graph shows cell number of MOLM-14, OCI-AML2, or OCI-AML3. Three days after transduction, cells were plated at 0.3 M cells/ml (D0) and cell proliferation was followed using trypan blue exclusion ($n = 4$ independent experiments for OCI-AML2 and $n = 6$ independent experiments for MOLM-14 and OCI-AML3, two-way ANOVA). (**b**) Graph shows the percentage of Annexin-V$^+$ or 7-AAD$^+$ cells 4 days after cell transduction ($n = 3$ independent experiments for MOLM-14 and OCI-AML3 and $n = 4$ independent experiments for OCI-AML2, unpaired $t$-test). (**c**) Investigation of the role of CALCRL on leukemic cell growth in vivo. **d** Leukemia burden measured using mCD45.1$^-$/hCD45$^+$/hCD33$^+$/AnnV$^-$ markers (unpaired $t$-test). **e** Mice survival monitoring (Log-rank (Mantel-Cox) test). **f** Investigation of the role of CALCRL on leukemic cell growth in vivo in a context of engrafted cells. **g** Measurement of Leukemia burden (unpaired $t$-test). **h** Mice survival monitoring (Log-rank (Mantel-Cox) test). Data are mean ± s.e.m.

shRNA (Fig. S3a) phenocopied the effects of shCALCRL on cell proliferation and apoptosis in MOLM-14 and OCI-AML3 cells (Fig. S3b, c). In order to confirm these results in vivo and to control the invalidation of the target over time, we have developed tetracycline-inducible shRNA models. First, we established that the inducible depletion of CALCRL was associated with a decrease in cell proliferation and an increase in apoptosis as observed with constitutive shRNA approaches (Fig. S3d–f). After injection of AML cells in mice, RNA depletion was activated from the first day by doxycycline (Fig. 2c). Twenty-five days post-transplantation, the engraftment of human leukemic cells from murine bone marrow and spleen was assessed with mCD45.1$^-$hCD45$^+$hCD33$^+$AnV$^-$ markers (Fig. S3g). Mice injected with shCAL#1 and shCAL#2 had a significant reduction in total cell tumor burden (as defined by AML blasts in the bone marrow and spleen) compared to shCTR (CTR for control) in both MOLM-14 (shCTR $= 13.9 \times 10^6$ cells vs shCAL#1 $= 0.3 \times 10^6$ cells vs shCAL#2 $= 0.1 \times 10^6$ cells) and OCI-AML3 cells (shCTR $= 17.2 \times 10^6$ cells vs shCAL#1 $= 2.0 \times 10^6$ cells vs

shCAL#2 $= 1.7 \times 10^6$ cells) (Figs. 2d and S3h). Finally, CALCRL silencing significantly prolonged mice survival (Fig. 2e). To take full advantage of our inducible constructs and to improve the clinical relevance of our model, we assessed the impact of CALCRL depletion on highly engrafting AML cells (Fig. 2f). Short hairpin RNA expression was induced 10 days post-transplantation of shCTR or shCAL in MOLM14 cells after verifying that the level of engraftment was similar in both groups (Fig. 2f, g). In this overt AML model, we observed a marked reduction in bone marrow blasts in the mice xenografted with shCAL AML cells compared to the shCTR-xenografted cohort in which the disease progressed (Fig. 2g). Furthermore, CALCRL downregulation significantly increased survival (Fig. 2g). Importantly, these results demonstrated that the reduction in blast number and the increased survival of mice observed after CALCRL depletion was not the consequence of an inhibition of leukemic blast homing to the murine bone marrow. Altogether, these results demonstrate that CALCRL is required both for the propagation and the maintenance of AML cells in vivo.

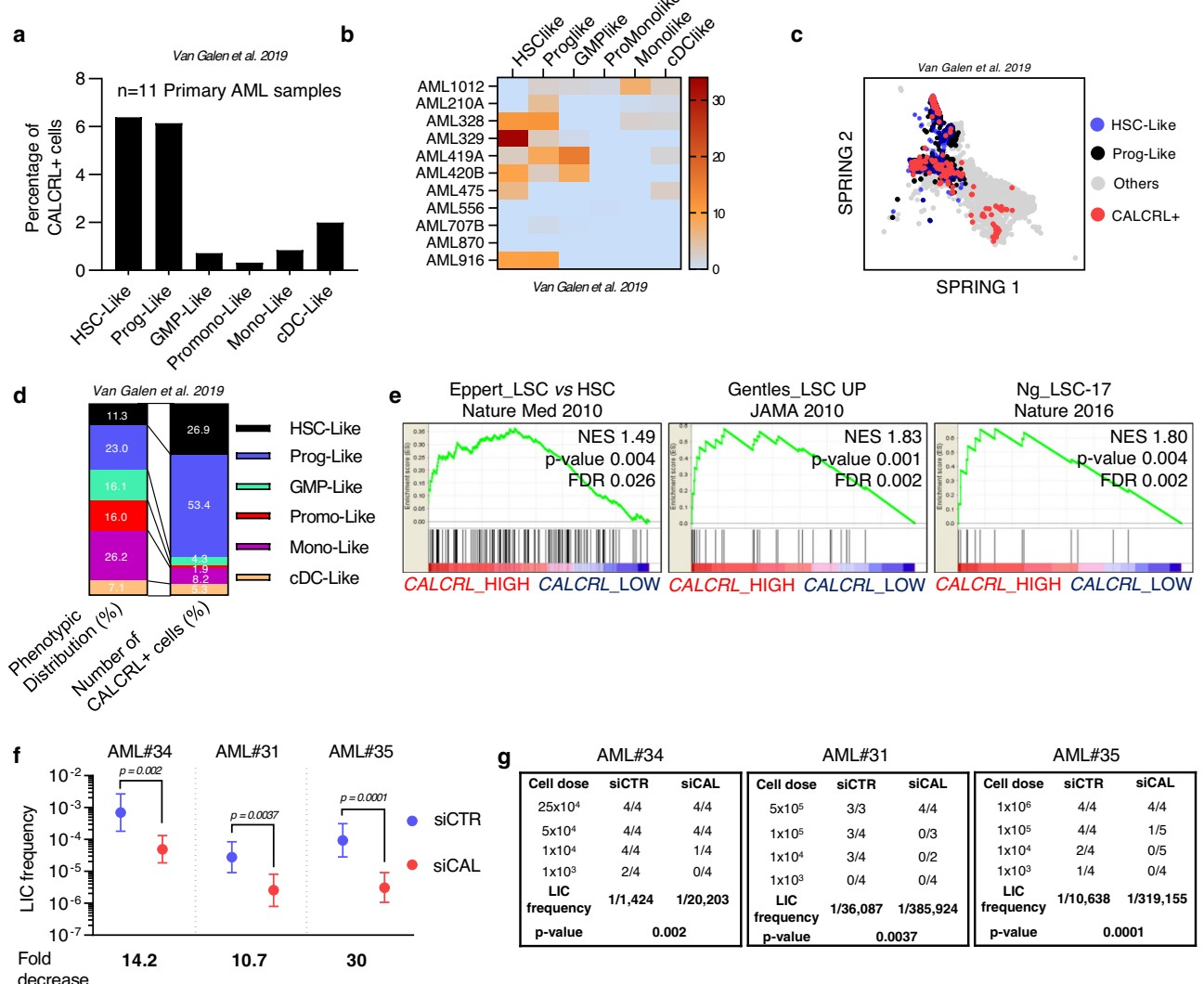

**Fig. 3 CALCRL is required for leukemic stem cell maintenance. a** Barplot shows the percentage of CALCRL[positive] cells in ($n = 11$ primary AML samples) in cells classified as HSC-like, progenitor-like, GMP-like, Promonocyte-like, Monocyte-like, or cDC-like malignant cells (Van Galen et al. 2019)[31]. **b** Heatmap shows the percentage of CALCRL[positive] cells in each patient individually. **c** SPRING visualization of single-cell transcriptomes. Points are color-coded by indicated cell-type annotations. **d** Phenotypic distribution and number of CALCRL[positive] cells. **e** GSEA of stem cell signatures functionally identified by Eppert et al.[2] or phenotypically defined by Gentles et al.[3] or Ng et al.[4] was performed using transcriptomes of cells expressing low (blue) vs high (red) levels of CALCRL gene (TCGA, AML cohort, Enrichment Score based on a Kolmogorov-Smirnov statistic). **f** Primary AML samples or cells from primary mice were collected and treated ex vivo with siCTR or siCALCRL and transplanted in limiting doses into primary or secondary recipients. Human marking of >0.1% was considered positive for AML engraftment except for AML#31 for which the cut-off was 0.5% because the sample was hCD33⁻ (Poisson statistic). **g** Engraftment results. Data are mean ± error bars (upper and lower limit, Poisson statistic).

**CALCRL is required for leukemic stem cell maintenance**. As *CALCRL* expression is linked to an immature phenotype and CALCRL-depletion impaired AML cell growth, we next aimed to address the role of CALCRL in LSC biology. First, we analyzed previously published single cell RNA-sequencing data[31] and observed that *CALCRL* is preferentially expressed in HSC-like and progenitor-like cells (Prog-like cells) compared to more committed cells in 11 AML patients (Fig. 3a–c). Moreover, while HSC-like and Prog-like cells represent only 34.3% of the total of leukemic cells found in these patients, they accounted for more than 80% of CALCRL[positive] cells (Fig. 3d). Gene set enrichment analysis (GSEA) confirmed that several LSC-associated gene signatures[2–4] (Table S4) are significantly enriched in AML patients (the Cancer Genome Atlas, AML cohort, 2013) exhibiting the highest *CALCRL* expression compared to AML patients with the lowest *CALCRL* expression (Fig. 3e). To functionally

investigate the role of CALCRL in LSC biology, we performed ex vivo assays knocking down CALCRL in 3 primary AML samples followed by limiting dilution assay (LDA). We demonstrated that in all the tested samples CALCRL inhibition significantly decreased the frequency of LSCs (Fig. 3f, g; see Fig. S4a for gating strategy), demonstrating the requirement of CALCRL in preserving the function of LSCs.

**Depletion of CALCRL alters cell cycle and DNA repair pathways in AML**. To examine regulatory pathways downstream of CALCRL, we generated and performed comparative transcriptomic and functional assays on shCTR vs shCALCRL MOLM-14 cells. CALCRL knockdown was associated with a significant decrease in the expression of 623 genes and an increase in 278 genes (FDR < 0.05) (Fig. 4a; see Data Source document). Data mining and western blotting analyses showed significant

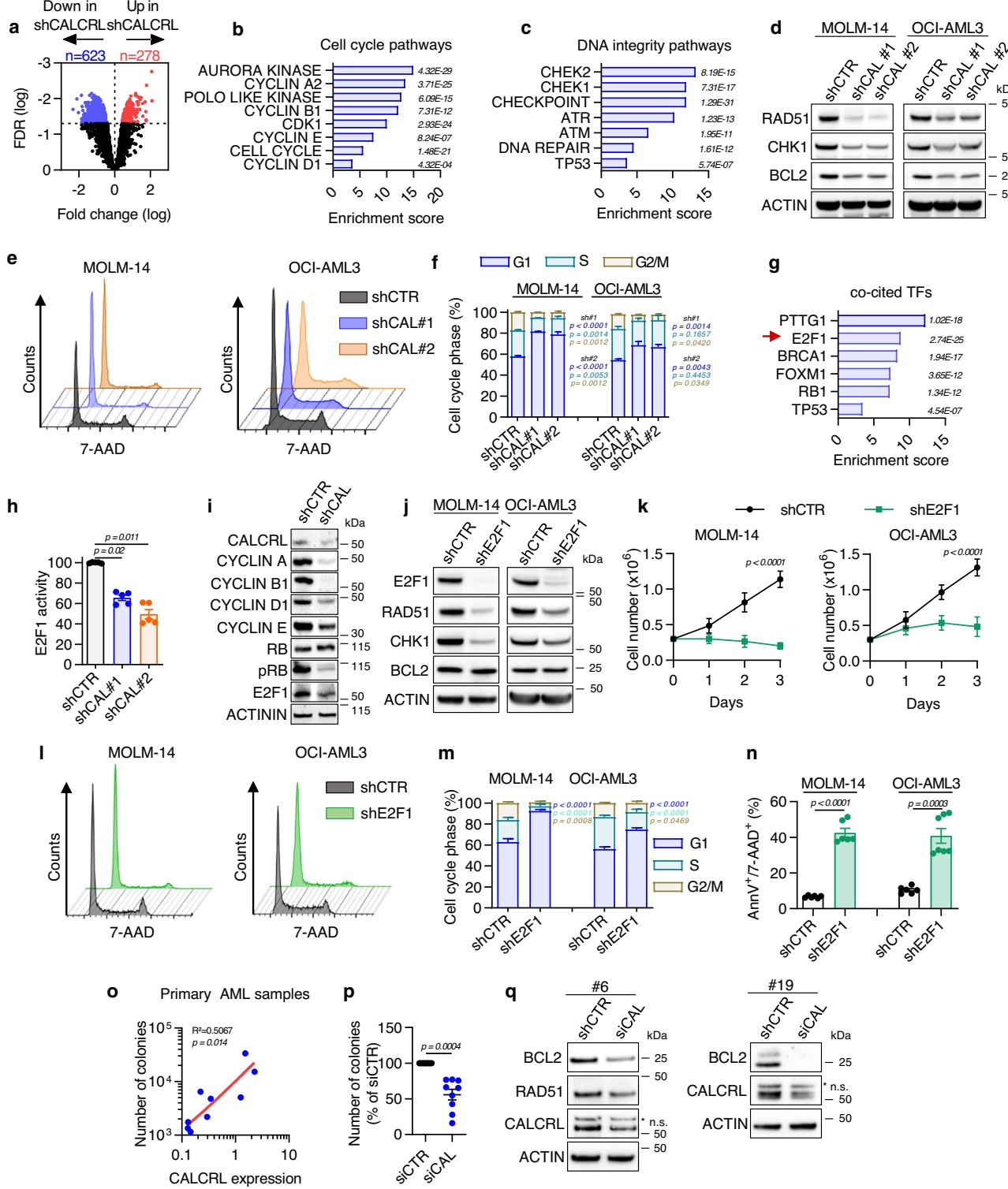

depletion in genes involved in cell cycle and DNA integrity pathways (Fig. 4b, c), and a reduction in protein level of RAD51, CHEK1, and BCL2 in shCALCRL AML cells (Fig. 4d). This was associated with an accumulation of cells in the $G_1$ phase (Fig. 4e, f). Interestingly, enrichment analysis showed that depletion of CALCRL affects the gene signatures of several key transcription factors such as E2F1, P53, or FOXM1 described as critical cell cycle regulators (Fig. 4g). We focused on the E2F1 transcription factor, whose importance in the biology of leukemic stem/progenitors cells has recently been shown[32]. We first confirmed that

CALCRL depletion was closely associated with a significant decrease in the activity of E2F1 (Fig. 4h). As E2F1 activity is mainly regulated by CDK-cyclin complexes, we assessed the protein expression of these actors after CALCRL downregulation. We observed a decrease in both the phosphorylation of Rb and the expression of cyclins A, B1, D1, and E, reflecting the cell cycle arrest in cells depleted for CALCRL (Fig. 4i).

Then, we demonstrated that the knockdown of E2F1 affected protein expression of RAD51, CHK1, but not BCL2 (Fig. 4j), inhibited cell proliferation (Fig. 4k), cell cycle progression (Fig. 4l, m),

**Fig. 4 CALCRL controls cell cycle and DNA repair pathways through E2F1. a** Volcano plot of most differentially expressed (278 upregulated and 623 downregulated, FDR < 0.05 and $p$-value < 0.05) genes identified in transcriptomes of MOLM-14 shCAL#1 and #2 vs shCTR. The FDR values based on −log10 were plotted against the log2 ratio of gene expression level for all genes. **b, c** Genomatix analysis of cell cycle (**b**) or DNA integrity (**c**) pathways negatively affected by CALCRL depletion. **d** Western blot results showing expression of RAD51, CHK1, BCL2, and β-ACTIN proteins in MOLM-14 and OCI-AML3 four days after transduction with indicated shRNA. **e** Representative FACS graphs of cell cycle upon shCAL. **f** Cell cycle distribution ($n = 3$ independent experiments, two-way ANOVA). **g** Genomatix analysis of co-cited transcription factors negatively affected by CALCRL depletion. **h** E2F1 activity is followed by flow cytometry through the level of mKate2 fluorescence intensity ($n = 5$ independent experiments, unpaired $t$-test). **i** Western blot results showing expression of Cyclin A, B1, D1, E, RB, and p-RB. **j** Western blot results showing expression of E2F1, RAD51, CHK1, BCL2, and β-ACTIN proteins in MOLM-14 and OCI-AML3 four days after transduction with indicated shRNA. **k** Graph shows cell proliferation of MOLM-14 or OCI-AML3. Three days after transduction, cells were plated at 0.3 M cells/ml (D0) and cell proliferation was followed using trypan blue exclusion ($n = 6$ independent experiments, two-way ANOVA). **l** Representative FACS graphs of cell cycle upon shE2F1. **m** Cell cycle distribution ($n = 5$ independent experiments for MOLM-14 and $n = 6$ independent experiments for OCI-AML3, two-way ANOVA). **n** Graph shows the percentage of Annexin-V + or 7-AAD + cells 4 days after cell transduction ($n = 6$ independent experiments for MOLM-14 and $n = 7$ independent experiments for OCI-AML3, unpaired $t$-test). **o** Correlation between clonogenic capacities of primary AML samples ($n = 11$) and CALCRL protein expression assessed by western blot analysis (Supplementary Fig. 1m). Linear regression was performed to determine $R^2$ and $p$-value. **p** Colonies in methylcellulose were counted 1 week after transfection of primary AML samples with siCTR or siCAL (unpaired $t$-test). **q** Western blot results showing expression of RAD51, BCL2, CALCRL, and β-ACTIN proteins in primary AML samples 7 days after transfection with indicated siRNA. Data are mean ± s.e.m.

and induced cell death in both MOLM-14 and OCI-AML3 (Fig. 4n). We further investigated whether CALCRL might regulate the proliferation of primary AML cells. Interestingly, CALCRL protein level positively correlated with clonogenic capacities in methylcellulose (Fig. 4o). Moreover, the depletion of CALCRL in primary samples decreased the number of colonies (Fig. 4p), and BCL2 and RAD51 protein levels (Fig. 4q). All these results suggest that CALCRL has a role in the proliferation of AML blasts and controls critical pathways involved in DNA repair processes.

**CALCRL downregulation sensitizes leukemic cells to chemotherapeutic drugs.** Based on putative targets of CALCRL such as BCL2, CHK1, or FOXM1[24,33,34], we hypothesized that CALCRL was involved in chemoresistance. Accordingly, CALCRL depletion sensitized MOLM-14 and OCI-AML3 cells to AraC and idarubicin as attested by the reduction in cell viability (Fig. 5a) and the induction of cell death (increased Annexin-V staining, Fig. 5b; and increased cleavage of apoptotic proteins Caspase-3 and PARP, Fig. S5a). Furthermore, depletion of ADM or E2F1 also sensitized AML cells to these genotoxic agents (Fig. S5b, c), demonstrating that the ADM-CALCRL-E2F1 axis was involved in chemoresistance in vitro. Importantly, siRNA-mediated depletion of CALCRL in seven primary AML samples combined with AraC significantly reduced the clonogenic growth of leukemic cells compared with siCTR+AraC and siCALCRL conditions (Fig. 5c). As we showed that CALCRL depletion affected DNA integrity pathways, we functionally investigated the role of CALCRL on the DNA repair by performing comet assays. We observed that both CALCRL and ADM shRNAs increased the length of comet tails and alkaline comet tail moments compared to control cells and that AraC significantly and further increased the alkaline comet tail moment in these cells (Fig. 5d, e). These results indicate that DNA repair pathways were affected when the ADM-CALCRL axis was impaired in AML.

**Downregulation of CALCRL sensitizes leukemic cells to chemotherapy in vivo.** We xenografted NSG mice with AML cell lines transduced with inducible shCALCRL demonstrating the same chemosensitization profile than constitutive shRNAs (Fig. S5d, e). After full engraftment, CALCRL was depleted by doxycycline and mice were treated with 30 mg/kg/day AraC for 5 days (Fig. 6a). While AraC alone had no effect on AML propagation, AraC in combination with shCALCRL significantly reduced the total number of blasts (Fig. 6b), induced a higher rate of cell death (Fig. 6c), and prolonged survival of mice (Fig. 6d)

compared to others conditions. Furthermore, MOLM-14 cells expressing shCTR and treated with vehicle or AraC were FACS-sorted and plated in vitro for further experiments. Interestingly, after 1 week of in vitro culture, human AML cells from AraC-treated mice were more resistant to AraC (IC$_{50}$: 1 µM for vehicle group vs 5.40 µM for AraC-treated group) and idarubicin (IC$_{50}$: 31.98 nM for vehicle group vs 111.3 nM for AraC treated group) (Fig. 6e). Next, we observed that AML cells treated with AraC in vivo had higher protein expression levels of CALCRL, and a slight increase in RAD51 and BCL2, whereas CHK1 was similar to untreated cells (Fig. 6f). To evaluate the role of CALCRL in this chemoresistance pathway in vivo, we depleted CALCRL in these cells. Knockdown of CALCRL by two different shRNAs sensitized cells to AraC and idarubicin compared to shCTR in cells treated with vehicle (Fig. 6g) or AraC alone (Fig. 6h). Remarkably, the IC$_{50}$ of AraC and idarucibin in AraC-treated cells in vivo and transduced with shCALCRL was also decreased, demonstrating that CALCRL participated to chemoresistance pathways in AML. We further aimed to determine what ligand might be involved into this chemoresistance pathway. First, we depleted RAMP1, RAMP2, or RAMP3 using shRNAs (Fig. S6a) and observed that only RAMP2 was necessary to sustain the growth of MOLM-14 cells (Fig. S6b). Consistent with this result, RAMP2 downregulation induced cell death and sensitized AML cell to AraC and Ida (Fig. S6c). We also showed that only exogenous ADM1-52 and ADM13-52, but not CGRP, were able to decrease AraC-induced cell death in MOLM-14 cell line, highlighting that the CALCRL-driven chemoresistance is mainly due to the ADM ligand (Fig. S6d). We used a PDX model treated with a CGRP inhibitor, olcegepant, in association with AraC to confirm these results (Fig. S7a). We showed that CGRP inhibition did not affect leukemic burden and did not sensitize to AraC in vivo (Fig. S7b). Interestingly, AraC upregulated CALCRL (Fig. S7c–e), confirming our in vitro observations in vivo in PDX model. As previously described[35] and as a positive control of the in vivo activity of olcegepant, we showed that this drug efficiently decreased the percentage of murine CMP but not LSK and GMP subpopulations (Fig. S7f–h). Altogether, these experiments indicate that ADM was the main ligand of CALCRL in AML.

Because mitochondrial metabolism has emerged as a critical regulator of cell proliferation and survival in basal and chemotherapy-treated conditions in AML[12,23,36–39], we analyzed the impact of CALCRL depletion on mitochondrial function. GSEA showed a significant depletion in the gene signature associated with mitochondrial oxidative metabolism in the shCALCRL MOLM-14 cells (Fig. S8a and Table S4). Mitochondrial oxygen consumption

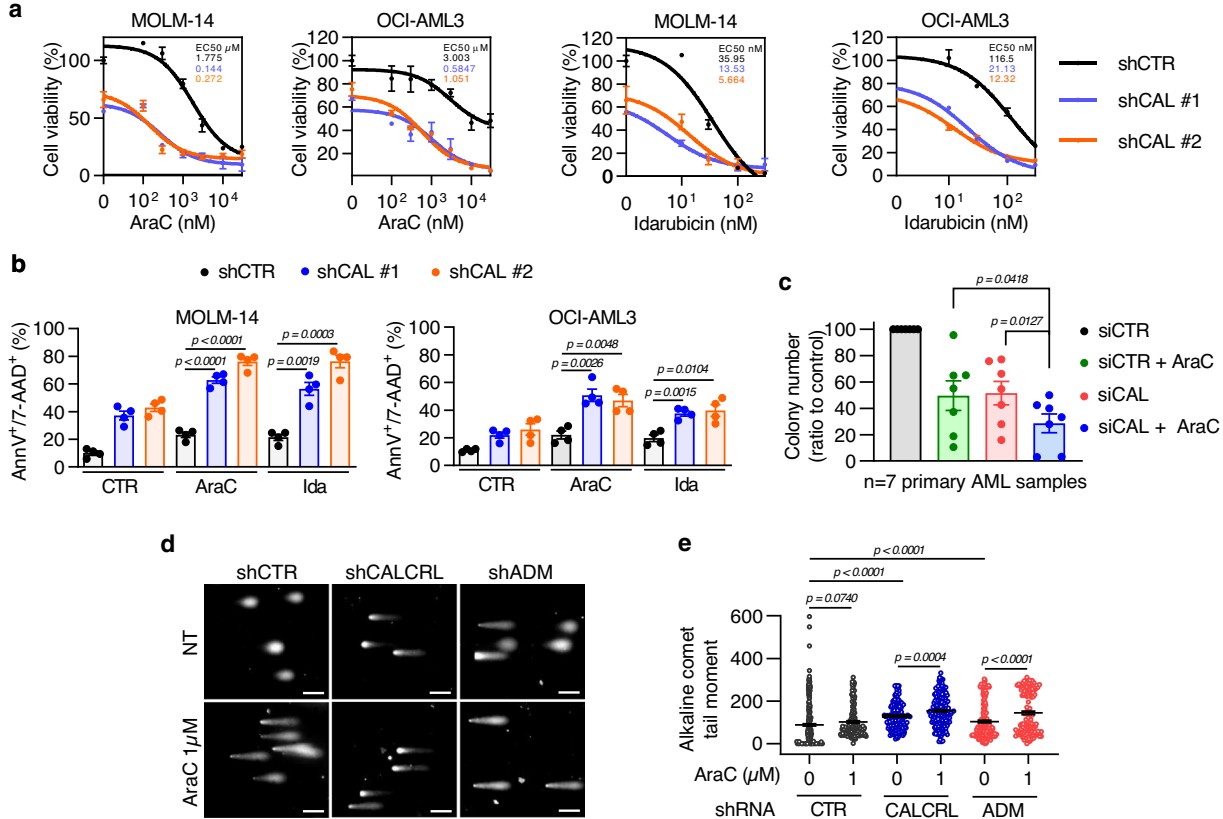

**Fig. 5 Depletion of CALCRL sensitizes AML cells to chemotherapy through induction of DNA damages. a** Four days after transduction, MOLM-14 or OCI-AML3 cells were treated with AraC or idarubicin for 48 h. Then cell viability was assessed by MTS assay and values were normalized to untreated condition ($n = 3$ independent experiments). Curve fit to calculate IC50 was determined by log (inhibitor) vs response (three parameters). **b** Graph shows the percentage of AnnV+ or 7-AAD+ cells. Four days after transduction, cells were treated with chemotherapeutic agents for 48 h before flow cytometry analysis ($n = 4$ independent experiments, unpaired $t$-test). **c** Colonies in methylcellulose were counted 1 week after transfection of primary AML samples with siCTR or siCAL+/- 5 nM AraC ($n = 7$ primary AML samples, paired $t$-est). **d** Detection of double strand breaks by alkaline comet assays in MOLM-14 cells untreated or treated 24 h with Ara-C (1 μM). Representative pictures of nuclei in the same experiment. **e** Quantification of alkaline comet tail moments for one representative experiment (135–413 nuclei were analyzed for each treatment, unpaired $t$-test) out of four. Data are mean ± s.e.m.

measurements revealed a modest but significant reduction in basal OCR, whereas maximal respiration was conserved, indicating that mitochondria remain functional (Fig. S8b). We also consistently observed a significant decrease in mitochondrial ATP production by shCALCRL (Fig. S8c). We and other groups have previously shown that chemoresistant cells have elevated oxidative metabolism and that targeting mitochondria in combination with conventional chemotherapy may represent an innovative therapeutic approach in AML[12,23,27]. Since depletion of CALCRL modestly decreased OCR and more greatly decreased mitochondrial ATP in AML cells (Fig. S8b, c), we assessed cellular energetic status associated with AraC. Knockdown of CALCRL significantly abrogated the AraC-induced increase in basal respiration and maximal respiration (Fig. S8d). Moreover, we observed a decrease in mitochondrial ATP production in response to AraC upon CALCRL silencing (Fig. S8e), whereas glycolysis (e.g., ECAR) was not affected (Fig. S8f).

As it has been reported that BCL2 controlled the oxidative status in AML cells[23], we investigated its role downstream of CALCRL. We showed that upon AraC treatment, the over-expression of BCL2 in MOLM-14 cells (Fig. S8g) is sufficient to rescue maximal respiration but not basal respiration. This suggested a role of the CALCRL-BCL2 axis in maintaining some aspects of mitochondrial function in response to AraC. This was not related to energy production, as neither mitochondrial ATP production nor ECAR were affected (Fig. S8i, j). Finally, BCL2 rescue almost entirely inhibited basal apoptosis induced by the

depletion of CALCRL and by the combination with AraC or idarubicin (Fig. S8k).

Overall, these results suggest that CALCRL mediates the chemoresistance of AML cells.

**Depletion of CALCRL in residual disease after AraC treatment impedes LSC function.** To address the role of CALCRL in response to chemotherapy in primary AML samples, we used a clinically relevant PDX model of AraC treatment in AML[12]. After engraftment of primary AML cells, NSG mice were treated for 5 days with AraC and sacrificed at day 8 to study the minimal residual disease (MRD; Fig. 7a). We analyzed 10 different PDX models and stratified them according to their response to AraC as low (fold change, FC AraC-to-Vehicle <10) or high (FC > 10) responders (Fig. 7b). The percentage of cells positive for CALCRL in the AML bulk was approximately doubled in the low responder group compared to the high responder group (3.6% vs 7.8%; Fig. 7c). We also observed an inverse correlation between the percentage of positive cells and the degree of tumor reduction ($p = 0.0434$; Fig. 7d). Moreover, after AraC treatment a significant increase in the percentage of blasts positive for CALCRL was observed (5.6% vs 23%; Fig. 7e) in all the CD34/CD38 sub-populations (Fig. 7f) from MRD. We next investigated the effects of AraC on ADM secretion. To this end, we evaluated the protein level of ADM in bone marrow supernatants of mice treated with

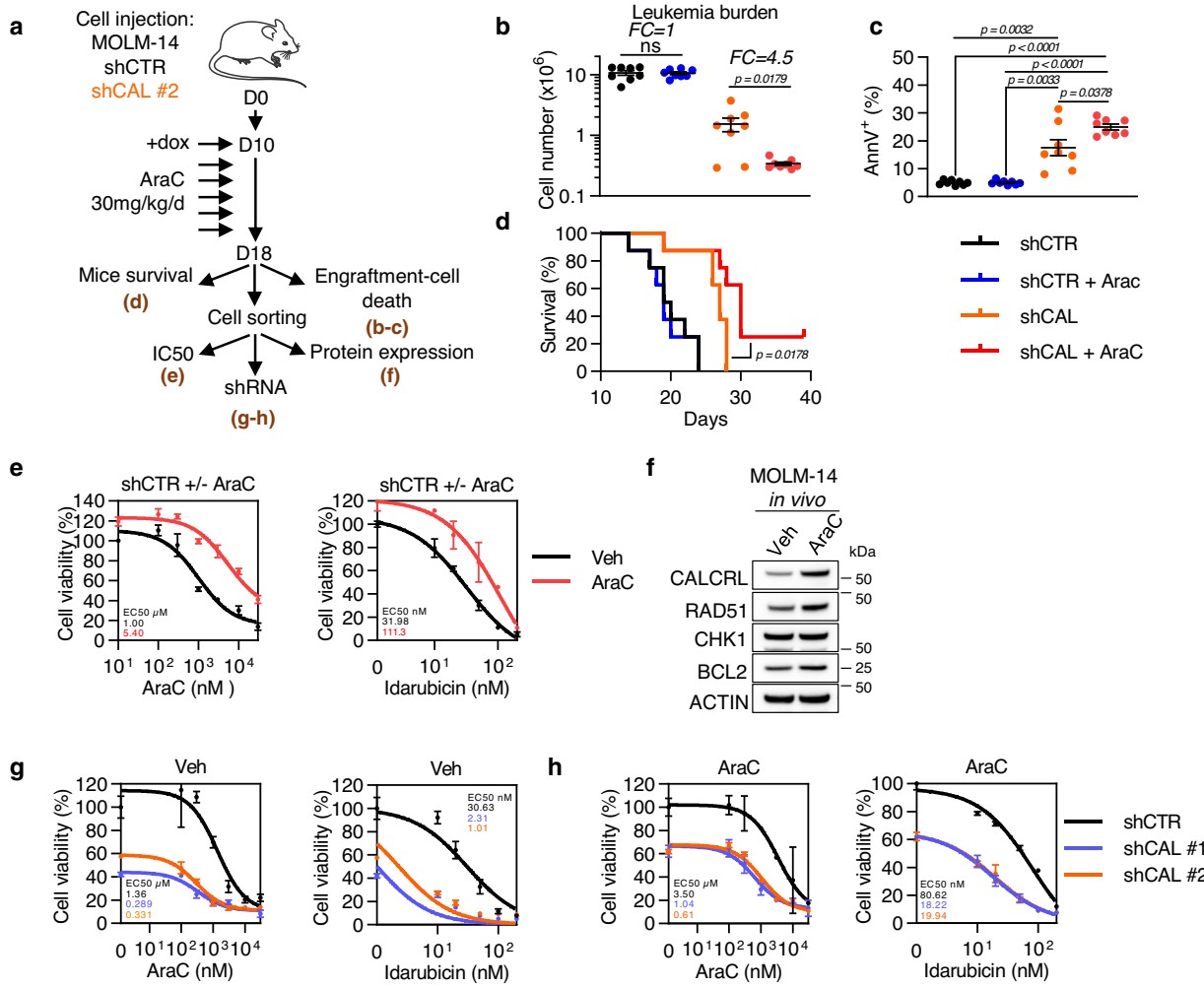

**Fig. 6 Depletion of CALCRL sensitizes AML cells to chemotherapy in vivo. a** Experimental plan for assessing the consequence of CALCRL depletion on chemotherapy response in vivo. $2 \times 10^6$ MOLM-14 expressing indicated inducible shRNAs were injected into the tail vein of NSG mice. Ten days later, when disease was established, mice were treated with 30 mg/kg/d AraC for 5 days. On day 18, a group of mice was kept to follow survival (**d**). The other group was sacrificed to assess human cell engraftment (**b**) and cell death (**c**). Human cells were also sorted by flow cytometry using mCD45.1$^-$/hCD45$^+$/hCD33$^+$/AnnV$^-$ markers and replated in culture dishes to assess protein expression (**f**), sensitivity to chemotherapeutic drugs (**e**), and to perform CALCRL depletion by shRNA (**g, h**). **b** Total leukemia burden measured using mCD45.1$^-$/hCD45$^+$/hCD33$^+$/AnnV$^-$ markers (unpaired $t$-test). **c** Graph shows the percentage of AnnV$^+$ cells (unpaired $t$-test). **d** Mice survival monitoring (Log-rank (Mantel-Cox) test). **e** After replating, cells were treated with AraC or idarubicin ex vivo for 48 h. Cell viability was assessed by MTS assay and values were normalized to untreated condition ($n = 1$ in quadriplate). Curve fit to calculate IC50 was determined by log (inhibitor) vs response (three parameters). **f** Western blotting for CALCRL, RAD51, CHK1, BCL2, and β-ACTIN. Cellular extracts were collected 2 days after mice sacrifice. **g, h** Human cells from vehicle (**g**) or AraC (**h**) treated mice were transduced with indicated shRNA. After 4 days cells were treated with AraC or idarubicin for 48 h and cell viability was assessed by MTS assay ($n = 1$ in quadriplate). Curve fit to calculate IC50 was determined by log (inhibitor) vs response (three parameters) test. Data are mean ± s.e.m.

PBS or AraC. Chemotherapy reduced both percentage of human cells (Fig. S9a) and levels of secreted ADM (Fig. S9b) in the bone marrow of mice. The correlation between leukemia burden and secretion of ADM reinforced the hypothesis of an autocrine secretion of ADM by leukemic blasts.

Next, we aimed at determining the role of CALCRL in the LSC function maintenance of the RIC population. Leukemic cells from patients collected at diagnosis were injected into NSG mice, and after engraftment and treatment with AraC, human viable AML cells constituting MRD were collected and transfected with siCTR or siCALCRL before LDA transplantations into secondary recipients (Fig. 7g). A significant reduction in LSC frequency was observed in the samples depleted from CALCRL compared to the controls in the two primary AML samples tested (Fig. 7h, i). We next examined cell surface expression of CALCRL in patients

before and after intensive chemotherapy (Fig. 7j). Treatment decreased the percentage of blasts in the bone marrow (Fig. 7k), accompanied with a significant enrichment in CALCRL$^{positive}$ cells (Fig. 7l). Moreover, we observed a continuous enrichment in CALCRL$^{positive}$ blasts following chemotherapy (12.9% at diagnosis, 32.8% at day 35, 81% at relapse; Fig. 7m). We further performed an ex vivo assay on a relapse sample followed by LDA in NSG mice (Fig. 7n). We demonstrated that CALCRL depletion significantly reduced LSC frequency, highlighting a critical role for CALCRL in the maintenance of the clone present at relapse (Fig. 7o, p). Recently Shlush et al.[40] proposed an elegant model of relapses with two situations: in the first one called "relapse origin-primitive" (ROp), relapse originated from rare LSC clones only detectable in HSPC or after xenotransplantation. In the second model, called "relapse origin-committed" (ROc), the relapse clone

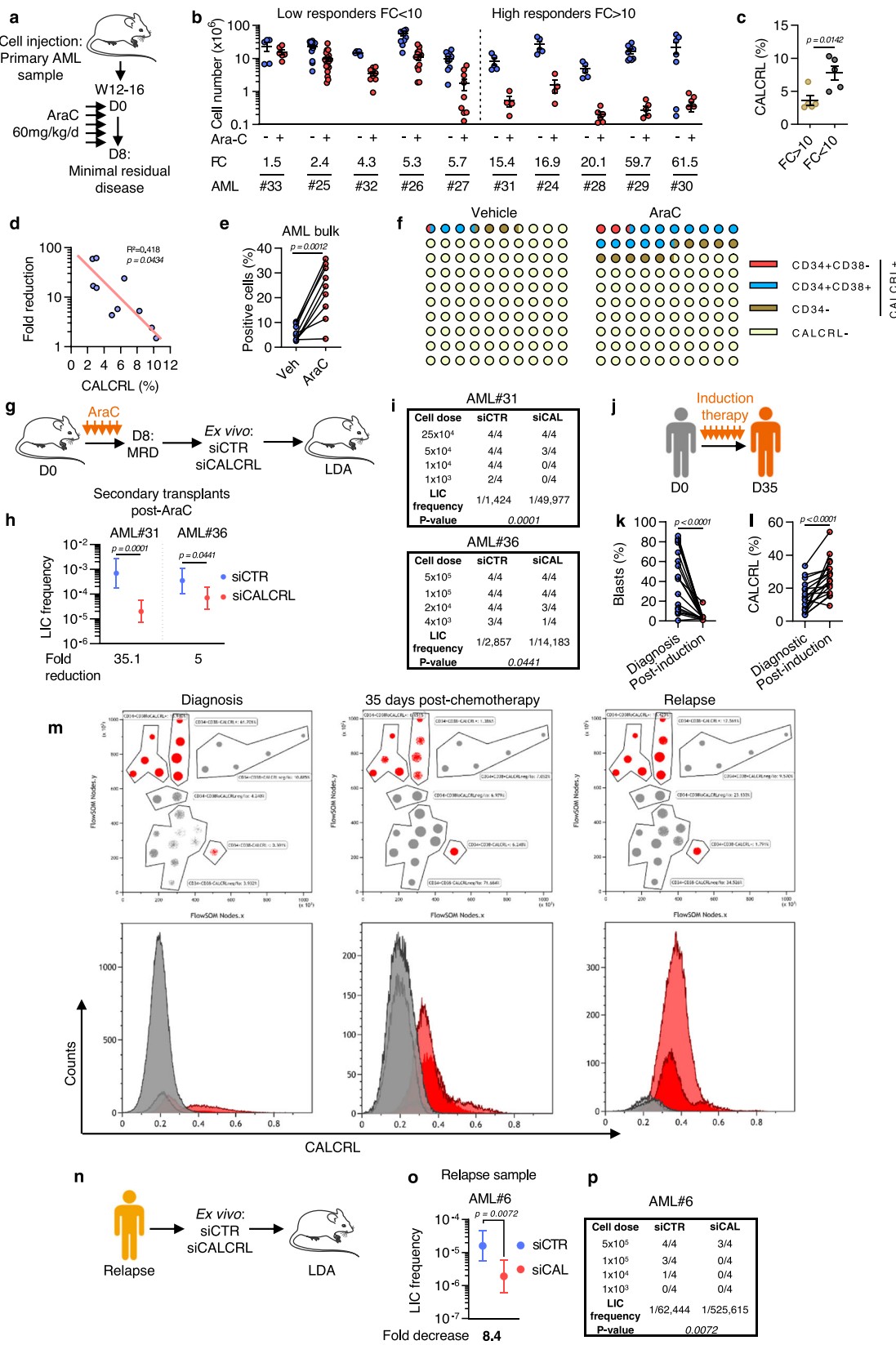

arose from immunophenotypically committed leukemia cells in which bulk cells harbored a stemness transcriptional profile[40]. We analyzed this transcriptomic database and found that at the time of diagnosis, *CALCRL* expression was higher in blasts with ROc than with ROp phenotype, in accordance with the expression of *CALCRL* in cells harboring stem cell features (Fig. S9C).

Interestingly, *CALCRL* was strongly increased at relapse in ROp patients, which correlated with the emergence of a clone with stem cell properties at this stage of the disease (Fig. S9d). These observations supported the hypothesis of the preexistence of a relapse-relevant LSC population, rare (ROp) or abundant (ROc), expressing high levels of *CALCRL*.

**Fig. 7 CALCRL predicts response to chemotherapy in PDX models and is required for the maintenance of RICs. a** Schematic diagram of the chemotherapy regimen and schedule used to treat NSG-based PDX models with AraC. **b** Total number of human viable AML cells. Patients were then separated into two categories: low responders (FC > 10) and high responders (FC > 10). **c** Graph shows the percentage of cells positive for CALCRL in low vs high responder groups (unpaired *t*-test). **d** Correlation between the fold reduction and the percentage of CALCRL[positive] cells (linear regression analysis). **e** Percentage of human CALCRL[positive] cells in vehicle-treated vs AraC-treated mice (unpaired *t*-test). **f** Picture representing the proportion of CALCRL[positive] cells before and after AraC treatment according to the studied cell population. **g** Role of CALCRL in the residual-LSC population. Human marking of >0.1% was considered positive for AML#36 sample and of 0.5% for AML#31 because the sample was hCD33[−]. **h** Graph shows the LIC frequency into bone marrow (Poisson statistic). **i** Engraftment results (Poisson statistic). **j** Monitoring of CALCRL expression at patient diagnosis or 35 days after intensive chemotherapy. **k** Percentage of blasts in patient bone marrow at diagnosis or after induction (paired *t*-test). **l** Percentage of human CALCRL[positive] cells in patients determined by flow cytometry (paired *t*-test). **m** Distribution of CALCRL[positive] and CALCRL[negative] cells at diagnosis, after chemotherapy, or at relapse. **n** Role of CALCRL in the LSC population at the time of relapse. **o** Graph shows the LIC frequency into bone marrow (Poisson statistic). **p** Engraftment results. Data are mean ± s.e.m or mean ± errors bars (upper and lower limit for LDA analysis, Poisson statistic).

Altogether, these results strongly support the conclusion that CALCRL preserves LSC function after chemotherapy, thus representing an attractive therapeutic target to eradicate the clone at the origin of relapse.

## Discussion

LSC-selective therapies represent an unmet need in AML due to high plasticity and heterogeneity not only of the phenotype[2,14,15], but also for the drug sensitivity[12,13] of the LSC population. However, fundamental studies focusing on intrinsic properties of LSCs such as their resistance to chemotherapy are crucially needed for the development of better and more specific therapies in AML.

Our study provides key insights of LSC biology and drug resistance and identifies the ADM receptor CALCRL as a master regulator of RICs. Our work first shows that *CALCRL* gene is overexpressed in the leukemic compartment compared to normal hematopoietic tissue based on Eppert's study that functionally characterizes LSCs. CALCRL could be specifically upregulated by LSC-related transcription factors such as HIF1α or ATF4[41,42]. Indeed, both *ADM* and *CALCRL* possess the consensus hypoxia-response element (HRE) in the 5′-flanking region and are HIF1α-regulated genes[43]. Recently, it has been demonstrated that the integrated stress response and the transcription factor ATF4 is involved in AML cell proliferation and is uniquely active in HSCs and LSCs[42,44]. Interestingly, maintenance of murine HSCs under proliferative stress but not steady-state conditions is dependent on CALCRL signaling[45]. Accordingly, CALCRL might support leukemic hematopoiesis and overcome stress induced by the high proliferation rate of AML cells. As opposed to ADM, CGRP is slightly expressed in BM from AML patient biopsies at diagnosis and its expression does not have a prognostic value in AML. Under inflammatory conditions to regulate immune responses or post-chemotherapeutic stress responses, continuous CGRP secretion is increased in normal hematopoietic cells or relapse-initiated cells as paracrine signal[35,45]. In this context, CGRP is one of the components of the CALCRL effects, while the ADM-CALCRL is the major driver of these RICs in an autocrine-dependent manner.

Our findings clearly show that targeting *CALCRL* expression impacts clonogenic capacities, cell cycle progression, and genes related to DNA repair and genomic stability. Cancer stem cells and LSCs are predominantly quiescent, thereby spared from chemotherapy. However, recent studies suggested that LSCs may also display a more active cycling phenotype[17,46]. C-type lectin CD93 is expressed on a subset of actively cycling, non-quiescent AML cells enriched for LSC activity[17]. Recently, Pei et al.[46] showed that targeting the AMPK-FIS1 axis disrupted mitophagy and induced cell cycle arrest in AML, leading to the depletion of LSC potential in primary AML. These results are consistent with the existence of different subpopulations of LSCs that differ in

proliferative state. Moreover, FIS1 depletion induces the down-regulation of several genes (e.g., CCND2, CDC25A, PLK1, CENPO, AURKB) and of the *E2F1* gene signature that were also identified after *CALCRL* knockdown. Recently, it has been proposed that E2F1 plays a pivotal role in regulating the CML stem/progenitor cells proliferation and survival status[32]. Several signaling pathways, for instance MAPKs, CDK/cyclin, or PI3K/AKT, have been described to be stimulated by ADM/CALCRL axis and may control pRB/E2F1 complex activity[47,48]. Other signaling mediators activated in LSCs such as c-Myc and CEBPα regulate E2F1 transcription and allow the interaction of the E2F1 protein with the E2F gene promoters to activate genes essential for DNA replication at G1/S, cell proliferation, and survival in AML[49–51]. Therefore, our analysis of cellular signaling downstream of CALCRL uncovers new pathways crucial for the maintenance and the chemoresistance of LSCs.

The characterization of RICs, which are detected at low threshold at diagnosis and strongly contributes to chemotherapy resistance, is necessary to develop new therapies with the aim of reducing AML relapses. Boyd and colleagues have proposed the existence of a transient state of LSCs during the immediate and acute response to AraC that is responsible for disease regrowth foregoing the recovery of LSC pool[13]. In this attractive model and in the dynamic period of post-chemotherapy MRD, CALCRL-positive AML cells belong to this leukemia-regenerating cell subpopulation and CALCRL is essential for the preservation of the LSC potential of primary chemoresistant AML cells. It would be interesting to determine whether chemotherapy rather primarily spares CALCRL-positive cells and/or induces an adaptive response that increases the expression of *CALCRL*.

In summary, while two distinct chemosensitive and chemoresistant LSC subpopulations coexist in AML, our study further defines this therapy-resistant LSC subpopulation responsible for relapse in patients and PDX as RICs. Our data also identify CALCRL as an AML actor of RICs. CALCRL is involved in chemoresistance mechanisms, and its depletion sensitizes AML cells to chemotherapy in vitro and in vivo. Finally, our results pinpoint CALCRL as a new and promising candidate therapeutic target for eradicating the LSC subpopulation that initiated relapse in AML.

## Methods

**Human studies**. De-identified primary AML patient specimens are from Toulouse University Hospital (TUH), Toulouse, France. Frozen samples of patients diagnosed with AML were obtained from TUH after signing a written informed consent for research use in accordance with the Declaration of Helsinki, and stored at the HIMIP collection (BB-0033-00060). According to the French law, HIMIP biobank collection has been declared to the Ministry of Higher Education and Research (DC 2008-307, collection 1) and obtained a transfer agreement for research applications (AC 2008-129) after approbation by our institutional review board and ethics committee (Comité de Protection des Personnes Sud-Ouest et Outremer II). Clinical and biological annotations of the samples have been declared to the CNIL (Comité National Informatique et Libertés, i.e., Data processing and

Liberties National Committee). See Table S5 for cytogenetics and mutation information on human specimens used in the current study.

**In vivo animal studies**. NSG (NOD.Cg-Prkdcscid Il2rgtm1WjI/SzJ) mice (Charles River Laboratories) were used for transplantation of AML cell lines or primary AML samples. Male or female mice ranging in age from 6 to 9 weeks were started on experiment and before cell injection or drug treatments, mice were randomly assigned to experimental groups. Mice were housed in sterile conditions using HEPA-filtered micro-isolators and fed with irradiated food and sterile water in the Animal core facility of the Cancer Research Center of Toulouse (France). All animals were used in accordance with a protocol reviewed and approved by the Institutional Animal Care and Use Committee of Région Midi-Pyrénées (France).

**Cell lines, primary cultures, and culture conditions**. For primary human AML cells, peripheral blood or bone marrow samples were frozen in FCS with 10% DMSO and stored in liquid nitrogen. The percentage of blasts was determined by flow cytometry and morphologic characteristics before purification. Cells were thawed in 37 °C water bath, washed in thawing media composed of IMDM, 20% FBS. Then cells were maintained in IMDM, 20% FBS, and 1% Pen/Strep (GIBCO) for all experiments.

Human AML cell lines were maintained in RPMI-media (Gibco) supplemented with 10% FBS (Invitrogen) in the presence of 100 U/ml of penicillin and 100 µg/ml of streptomycin, and were incubated at 37 °C with 5% $CO_2$. The cultured cells were split every 2–3 days and maintained in an exponential growth phase. All AML cell lines were purchased at DSMZ or ATCC, and their liquid nitrogen stock were renewed every 2 years. These cell lines have been routinely tested for Mycoplasma contamination in the laboratory. The U937 cells were obtained from the DSMZ in February 2012 and from the ATCC in January 2014. MV4-11 and HL-60 cells were obtained from the DSMZ in February 2012 and 2016, respectively. KG1 cells were obtained from the DSMZ in February 2012 and from the ATCC in March 2013. KG1a cells were obtained from the DSMZ in February 2016. MOLM14 was obtained from Pr. Martin Carroll (University of Pennsylvania, Philadelphia, PA) in 2011.

**Mouse xenograft model**. NSG mice were produced at the Genotoul Anexplo platform at Toulouse (France) using breeders obtained from Charles River Laboratories. Transplanted mice were treated with antibiotic (Baytril) for the duration of the experiment. For experiments assessing the response to che-motherapy in PDX models, mice (6–9 week old) were sublethally treated with busulfan (30 mg/kg) 24 h before injection of leukemic cells. Leukemia samples were thawed in 37 °C water bath, washed in IMDM 20% FBS, and suspended in Hank's balanced salt solution at a final concentration of 1–10 × 10^6 cells per 200 µl for tail vein injection in NSG mice; 8–18 weeks after AML cell transplantation and when mice were engrafted (tested by flow cytometry on peripheral blood or bone marrow aspirates), NSG mice were treated by daily intraperitoneal injection of 60 mg/kg AraC or vehicle (PBS) for 5 days. AraC was kindly provided by the pharmacy of the TUH. Mice were sacrificed at day 8 to harvest human leukemic cells from murine bone marrow. For AML cell lines, mice were treated with busulfan (20 mg/kg) 24 h before injection of leukemic cells. Then cells were thawed and washed as previously described, suspended in HBSS at a final concentration of 2 × 10^6 per 200 µl before injecting into the bloodstream of NSG mice. For experiments using inducible shRNAs, doxycycline (0.2 mg/ml + 1% sucrose; Sigma-Aldrich, Cat# D9891) was added to drinking water, the day of cell injection or 10 days after until the end of the experiment. Mice were treated by daily intraperitoneal injection of 30 mg/kg AraC for 5 days and sacrificed at day 8. Daily monitoring of mice for symptoms of disease (ruffled coat, hunched back, weakness, and reduced mobility) determined the time of sacrificing for injected animals with signs of distress.

**Assessment of leukemic engraftment**. At the end of the experiment, NSG mice were humanely killed in accordance with European ethics protocol. Bone marrow (mixed from tibias and femurs) and spleen were dissected and flushed in HBSS with 1% FBS. MNCs from bone marrow and spleen were labeled with anti-hCD33, anti-mCD45.1, anti-hCD45, anti-hCD3, and/or anti-hCD44 (all from BD) anti-bodies to determine the fraction of viable human blasts (hCD3−hCD45+mCD45.1− hCD33+/hCD44+AnnV− cells) using flow cytometry. In some experiments, we also added anti-CALCRL, anti-CD34, and anti-CD38 to characterize AML stem cells. Monoclonal antibody recognizing extracellular domain of CALCRL was generated in the lab with the help of Biotem (France). Then antibody was labeled with R-phycoerythrin using Lightning-Link kit (Expedeon). All antibodies were used at concentrations between 1/50 and 1/200 depending on specificity and cell density. Acquisitions were performed on a LSRFortessa flow cytometer with DIVA software v6.1.2 (BD Biosciences) or CytoFLEX flow cytometer with CytoExpert software v2.0 (Beckman Coulter), and analyses with Flowjo v10.4.2. The number of AML cells/µl peripheral blood and number of AML cells in total leukemia burden (in bone marrow and spleen) were determined by using CountBright beads (Invitrogen) using the described protocol.

For LDA experiments, human engraftment was considered positive if at least >0.1% of cells in the murine bone marrow were hCD45+mCD45.1−hCD33+.

The cut-off was increased to >0.5% for AML#31 because the engraftment was measured only based on hCD45+mCD45.1−. Limiting dilution analysis was performed using L-calc software. List of antibodies used in this work: Anti-CALCRL (Jean-Emmanuel Sarry lab), Anti-mCD45.1 PERCPCY5.5 (BD, Cat# 560580), Anti-hCD45 APC (BD, Cat# 555485), Anti-hCD34 AF700 (BD, Cat# 561440), Anti-hCD38 PECY7 (BD, Cat# 335825), Anti-hCD33 PE (BD, Cat# 555450), and Anti-hCD44 BV421 (BD, Cat# 562890).

**Immunochemistry analysis**. Protein expression was investigated by immunor-eactivity scoring on tissue microarrays containing pre-therapeutic bone marrow samples from intensively treated AML patients. Studies on the tissue microarray have been approved by the institutional review board of the University of Münster. Detailed information on the AML tissue microarray cohort and CALCRL expression has been published previously (Angenendt et al.)[52]. AML tissue microarrays were stained using an anti-ADM (Abcam, ab69117) antibody as described previously (Angenendt et al.)[53]. Briefly, following deparaffinization and heat-induced epitope unmasking, 4 µm tissue sections were incubated with the primary antibodies, followed by suitable secondary and tertiary antibodies (Dako). Immunoreactions were visualized with a monoclonal APAAP-complex and a fuchsin-based substrate-chromogen system (Dako). Counterstaining was per-formed with Mayer's hemalum (Merck). Two investigators who were blinded towards patient characteristics and outcome independently assessed intensity of staining (1 = no/weak, 2 = moderate, 3 = strong staining intensity) and percentage of stained blasts at each intensity level. Subsequently, H-scores were calculated as described previously [H-score = 1 × (percentage of blasts positive at 1) + 2 × (percentage of blasts positive at 2) + 3 × (percentage of blasts positive at 3)] (Angenendt et al.)[52]. There was a good inter-investigator agreement ($r = 0.91$ for ADM, $p < 0.0001$). Samples from 179 AML patients were evaluable for CALCRL and ADM. Images were taken using a Nikon Eclipse 50i microscope equipped with a Nikon DS-2Mv.

**Western blot analysis**. Proteins were resolved using 4–12% polyacrylamide gel electrophoresis Bis-Tris gels (Life Technology, Carlsbad, CA) and electro-transferred to nitrocellulose membranes. After blocking in Tris-buffered saline (TBS) 0.1%, Tween 20%, and 5% bovine serum albumin, membranes were immunostained overnight with appropriate primary antibodies followed by incu-bation with secondary antibodies conjugated to HRP. Immunoreactive bands were visualized by enhanced chemiluminescence (ECL Supersignal West Pico; Thermo Fisher Scientific) with a Syngene camera. Quantification of chemiluminescent signals was done with the GeneTools software v4.3.8.0 from Syngene. List of antibodies used in this work: anti-CASPASE-3 (CST, Cat#9662; 1/1000), anti-ACTIN (Millipore, Cat# MAB1501; 1/10,000), anti-CALCRL (Elabscience, Cat# ESAP13421; 1/1000), anti-RAD51 (Abcam, Cat# ab133534; 1/1000), anti-BCL2 (CST, Cat# 2872; 1/1000), anti-E2F1 (C-20)(Santa Cruz, Cat# sc-193; 1/1000), anti-CHK1 (Santa Cruz, Cat# sc-8408; 1/1000), anti-RAMP1 (3B9) (Santa Cruz, Cat# sc-293438; 1/1000), anti-RAMP2 (B-5) (Santa Cruz, Cat# sc-365240; 1/1000), anti-RAMP3 (G-1) (Santa Cruz, Cat# sc-365313; 1/1000), anti-ADM (Thermo Fisher Scientific, Cat# PA5-24927; 1/1000), anti-CGRP (Abcam, Cat# ab47027; 1/1000), anti-PARP (Thermo Fisher Scientific, Cat# 44-698G; 1/1000), and anti-alpha/beta-Tubulin (CST, Cat# 2148; 1/1000).

**Cell death assay**. After treatment, $5 \times 10^5$ cells were washed with PBS and resuspended in 200 µl of Annexin-V binding buffer (BD, Cat# 556420). Two microliters of Annexin-V-FITC (BD, Cat# 556454) and 7-amino-actinomycin D (7-AAD; Sigma Aldrich) were added for 15 min at room temperature in the dark. All samples were analyzed using LSRFortessa or CytoFLEX flow cytometer.

**Cell cycle analysis**. Cells were harvested, washed with PBS, and fixed in ice-cold 70% ethanol at −20 °C. Cells were then permeabilized with 1× PBS containing 0.25% Triton X-100, resuspended in 1× PBS containing 10 µg/ml propidium iodide and 1 µg/ml RNase, and incubated for 30 min at 37 °C. Data were collected on a CytoFLEX flow cytometer.

**Clonogenic assay**. Primary cells from AML patients were thawed and resuspended in 100 µl Nucleofector Kit V (Amaxa, Cologne, Germany). Then, cells were nucleofected according to the manufacturer's instructions (program U-001 Amaxa, Cologne, Germany) with 200 nM siRNA scrambled (ON-TARGETplus Non-targeting siRNA #2, Dharmacon) or anti-CALCRL (SMARTpool ON-TARGETplus CALCRL siRNA, Dharmacon). Cells were adjusted to $1 \times 10^5$ cells/ml final concentration in H4230 methylcellulose medium (STEMCELL Technol-ogies) supplemented with 10% 5637-CM as a stimulant and then plated in 35-mm petri dishes in duplicate and allowed to grow for 7 days in a humidified $CO_2$ incubator (5% $CO_2$, 37 °C). At day 7, the leukemic colonies (>5 cells) were scored.

**Plasmid cloning, shRNA, lentiviral production, and leukemic cell transduction**. shRNA sequences were constructed into pLKO-TET-ON or bought cloned into pLKO vectors. Each construct (6 µg) was co-transfected using lipofectamine 2000 (20 µl) in 10 cm dish with psPax2 (4 µg, provides packaging proteins) and pMD2.G (2 µg,

provides VSV-g envelope protein) plasmids into 293T cells to produce lentiviral particles. Twenty-four hours after cell transfection, medium was removed and 10 ml opti-MEM + 1% Pen/Strep was added. At about 72 h post transfection, 293T culture supernatants containing lentiviral particles were harvested, filtered, aliquoted, and stored at −80 °C in a freezer for future use. On the day of transduction, cells were infected by mixing $2 \times 10^6$ cells in 2 ml of freshly thawed lentivirus and Polybrene (Sigma-Aldrich, Cat# 107689) at a final concentration of 8 µg/ml. At 3 days post infection, transduced cells were selected using 1 µg/ml puromycin. List of plasmids used in this work: pCDH-puro-Bcl2 (Cheng et al.[54], Addgene plasmid #46971), Tet-pLKO-puro (Wiederschain et al.[55], Addgene plasmid #21915), CALCRL MISSION shRNA (shCALCRL#1, Sigma-Aldrich, Cat TRCN0000356798; shCALCRL#2, Sigma-Aldrich, Cat# TRCN0000356736), and E2F1 MISSION shRNA (Sigma-Aldrich, Cat# TRCN0000039659). List of shRNA sequences: shCALCRL#1 FW CTTATCTCGCTTG GCATATTC, shCALCRL#2 FW TTACCTGATGGGCTGTAATTA, shE2F1 FW CGC TATGAGACCTCACTGAAT, shRAMP1 FW CCCTTCTTCCAGCCAAGAAGA, shRAMP2 FW GAGCTTCTCAACAACCATGTT, and shRAMP3 FW GGACTAG GACTCCTTGCTTGA.

**IC$_{50}$ experiments**. The day before experiment, cells were adjusted to $3 \times 10^5$ cells/ml final concentration and plated in a 96-well plate (final volume: 100 µl). To measure half-maximal inhibitory concentration (IC$_{50}$), increased concentrations of AraC or idarubicin were added to the medium. After 2 days, 20 µl/well of MTS solution (Promega) was added for 2 h and then absorbance was recorded at 490 nm with a 96-well plate reader. The doses that decrease cell viability to 50% (IC$_{50}$) were analyzed using nonlinear regression log (inhibitor) vs response (three parameters) with GraphPad Prism (v6 and v8) software.

**Measurement of oxygen consumption in AML cultured cells using Seahorse assay**. All XF assays were performed using the XFe24 Extracellular Flux Analyser (Seahorse Bioscience, North Billerica, MA). The day before the assay, the sensor cartridge was placed into the calibration buffer medium supplied by Seahorse Biosciences to hydrate overnight. Wells of Seahorse XFe24 microplates were coated with 50 µl of Cell-Tak (Corning; Cat#354240) solution at a concentration of 22.4 µg/ml and kept at 4 °C overnight. Then, Cell-Tak-coated Seahorse microplates were rinsed with distilled water and AML cells were plated at a density of $10^5$ cells/well with XF base minimal DMEM media containing 11 mM glucose, 1 mM pyruvate, and 2 mM glutamine. Then, 180 µl of XF base minimal DMEM medium was placed into each well and the microplate was centrifuged at 80$g$ for 5 min. After 1 h incubation at 37 °C in CO$_2$-free atmosphere, basal oxygen consumption rate (OCR, as a mitochondrial respiration indicator), and extracellular acidification rate (ECAR, as a glycolysis indicator) were performed using the Seahorse XFe24, and analyzed using Wave software (version 2.6.1).

**Alkaline comet assays**. Alkaline comet assays were performed with OxiSelect Comet Assay Kit and according to the manufacturer's instructions (Cell Biolabs Inc.). Electrophoresis was performed at 4 °C in alkaline condition at 20 V during 45 mn. Slides were visualized by using a fluorescence microscope (AxioObserver Z1; Zeiss). Comet tail moments were measured with ImageJ software (version v1.8.0) with the plugin OpenComet v1.3.1 (http://opencomet.org/). Apoptotic cells were excluded from the analysis.

**RNA microarray and bioinformatics analyses**. For primary AML samples, human CD45$^+$ CD33$^+$ were isolated using cell sorter cytometer from engrafted BM mice (for 3 primary AML specimens) treated with PBS or with AraC. RNA from AML cells was extracted using Trizol (Invitrogen) or RNeasy (Qiagen). For MOLM-14 AML cell line, mRNA from $2 \times 10^6$ of cells was extracted using RNeasy (Qiagen). RNA purity was monitored with NanoDrop 1ND-1000 spectro-photometer and RNA quality was assessed through Agilent 2100 Bionalyzer with RNA 6000 Nano assay kit. No RNA degradation or contamination were detected (RIN > 9). Of the total RNA, 100 ng was analyzed on Affymetrix GeneChip© Human Gene 2.0 ST Array using the Affymetrix GeneChip© WT Plus Reagent Kit according to the manufacturer's instructions (Manual Target Preparation for GeneChip® Whole Transcript (WT) Expression Arrays P/N 703174 Rev. 2). Arrays were washed and scanned; and the raw files generated by the scanner was transferred into R software v4.0 for preprocessing (with RMA function, Oligo package), quality control (boxplot, clustering, and PCA), and differential expression analysis (with eBayes function, LIMMA package). Prior to differential expression analysis, all transcript clusters without any gene association were removed. Mapping between transcript clusters and genes were done using annotation provided by Affymetrix (HuGene-2_0-st-v1.na36.hg19.transcript.csv) and the R/Bioconductor package hugene20sttranscriptcluster.db. The $p$-values generated by eBayes function were adjusted to control false discovery using the Benjamini and Hochberg's procedure. [RMA] Irizarry et al.[56]; [Oligo package] Carvalho and Irizarry[57]; [LIMMA reference] Ritchie et al.[58]; hugene20sttran-scriptcluster.db: MacDonald JW 2017[59], Affymetrix hugene20 annotation data (chip hugene20sttranscriptcluster); [FDR]: Benjamini et al., Journal of the Royal Statistical Society, 1995[60].

**GSEA analysis**. GSEA analysis was performed using GSEA version 4.0 (Broad Institute). Gene signatures used in this study were from Broad Institute database, literature, or in-house built. Following parameters were used: Number of permutations = 1000, permutation type = gene_set. Other parameters were left at default values.

**Statistical analysis**. We assessed the statistical analysis of the difference between two sets of data using two-tailed (non-directional) Student's $t$-test with Welch's correction. For survival analyses, we used Log-rank (Mantel-Cox) test. Analyses were performed using Graphpad Prism (v6 and v8). For LDA experiments, frequency and statistics analyses were performed using L-calc software (Stemcell Technologies). A $p$ value of less than 0.05 indicates significance. $^*p < 0.05$; $^{**}p < 0.01$; $^{***}p < 0.001$; $^{****}p < 0.0001$; ns, not significant. Detailed information of each test is in the figure legends.

**Reporting summary**. Further information on research design is available in the Nature Research Reporting Summary linked to this article.

## Data availability

All data are available from the authors upon request. Source data are provided with this paper for Figs. 1–7 and Supplementary Figs. 1–9. RNA microarray dataset on shControl and shCALCRL MOLM14 cells in independent triplicates were deposited at GEO with accession code GSE162628. All publicly accessible transcriptomic databases of AML patients used in this study: GSE30377: Eppert et al.[2] GSE14468: Verhaak et al.[61]. GSE12417: Metzeler et al.[62]. GSE116256: Van Galen et al.[31] TCGA: The Cancer Genome Atlas Research Network[63]. BEATAML: Tyner ey al.[64]. Source data are provided with this paper.

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

## Acknowledgements

We thank all members of mice core facilities (UMS006, ANEXPLO, Inserm, Toulouse) for their support and technical assistance, and Prof. Véronique De Mas and Eric Delabesse for the management of the Biobank BRCHIMIP (Biological Resources Centres-INSERM Midi-Pyrénées "Cytothèque des hémopathies malignes") that is supported by CAPTOR (Cancer Pharmacology of Toulouse-Oncopole and Région). We thank the cellular ima- ging facility of U1037-CRCT. We thank the flow cytometry core facilities of U1037-CRCT and U1048-I2MC for technical assistance, and Anne-Marie Benot, Muriel Serthelon and Stéphanie Nevouet for their daily help with the administrative and financial management of our Team RESISTAML. This work was supported by grants from the French gov- ernment under the program "Investissement d'avenir" CAPTOR (ANR-11-PHUC-001), the Labex TOUCAN, the Fondation Toulouse Cancer Santé, the Fondation ARC, the Ligue National de Lutte Contre le Cancer, and from the Association G.A.E.L.

## Author contributions

Conception and design: C.L., J.-E.S. Development of methodology: C.L., J.E.S., V.F., C.R., J.T., T.K., C.S. Acquisition of data (provided animals, acquired and managed AML samples, provided facilities, etc.): C.L., M.D., P.L.M., T.F., M.G., C.B., E.S., M.-L.N.-T., M.S., N.S., Q.H., N.A., L.S., T.K., L.A., J.-H.M., S.M., N.G., E.B, A.S. Analysis and interpretation of data: C.L., J.-E.S. Study supervision: J.-E.S.

## Competing interests

The authors declare no competing interests.
