## [Peer Review File · Nature Communications]

REVIEWERS' COMMENTS

Reviewer #1 (Remarks to the Author):

The authors have addressed all my concern and added new data as well as revised the text. The revised version has much improved and teh conclusion have been srengthened and in my view is now ready to be published.

Reviewer #2 (Remarks to the Author):

The authors have performed many experiments that further confirm the role of Adrenomedullin-CALCRL Axis in human leukaemia biology. this is not being contested. Despite this additional work, the author have provided no additional novelty (as indicated oriingally from their own work and others), and alternatively only strengthened the original, but not novel findings.

Furthermore, the inability to compare signatures with the related, and very similar work of Farge et al. and Boyd et al, but continue to term these cells RIC, and not LRCs, or use gene expression signatures already published from these studies remains concerning, and only confuses the field further, and underscores the lack of novelty here. Other than discussion these points, the study has improved, but does not serve to advance the field beyond what is already in the literature. Publication in a specialized journal of this improved work will like reach and benefit the focused, and correct audience eg, role of ramps etc, in AML biology as determined in cell line surrogates.

Reviewer #3 (Remarks to the Author):

The response to the critiques is satisfactory and the paper is improved. one point remains- for western blots, the protein loading control should be done on the same gel. The authors are requested to repeat the immunblot and do the controls on the same gel.

Adrenomedullin-CALCRL Axis Controls Relapse-Initiating Drug Tolerant Acute Myeloid Leukemia Cells

We thank the reviewers for their constructive and thoughtful comments concerning our revised manuscript. Here we answer all comments and concerns of the reviewers as outlined below and we hope that the revised manuscript is now suitable for transfer and publication in *Nature Communications*.

Answers to reviewer's comments from the authors

Reviewer #1 (Remarks to the Author):

The authors have addressed all my concern and added new data as well as revised the text. The revised version has much improved and the conclusion have been strengthened and in my view is now ready to be published.

We thank the reviewer.

Reviewer #2 (Remarks to the Author):

The authors have performed many experiments that further confirm the role of Adrenomedullin-CALCRL Axis in human leukaemia biology. this is not being contested. Despite this additional work, the author have provided no additional novelty (as indicated originally from their own work and others), and alternatively only strengthened the original, but not novel findings.

Furthermore, the inability to compare signatures with the related, and very similar work of Farge et al. and Boyd et al, but continue to term these cells RIC, and not LRCs, or use gene expression signatures already published from these studies remains concerning, and only confuses the field further, and underscores the lack of novelty here. Other than discussion these points, the study has improved, but does not serve to advance the field beyond what is already in the literature. Publication in a specialized journal of this improved work will like reach and benefit the focused, and correct audience eg, role of ramps etc, in AML biology as determined in cell line surrogates.

We agree with the reviewer that we need to clarify the terms used for the cell populations studied in our work. This work focused on the relapse-initiating cells (RICs) defined as drug resistant leukemic stem cell (LSC) population responsible for relapse in patients and PDX.

As highlighted in our introduction, recent works (our study in 2017 and Boyd et al in 2018) have demonstrated that the anti-AML chemotherapy cytarabine (AraC) might deplete the LSC pool in patient-derived xenograft models. These results suggest the coexistence of two distinct LSC populations, one chemosensitive and thus eradicated by conventional treatments, and one that is chemoresistant, persists and might induce relapse in AML that we have defined as RICs. Boyd and colleagues have named them as leukemia-regenerating cells (LRCs).

We thought that the abbreviation of RICs more accurately reflects this therapy-resistant LSC subpopulation that we have characterized through the CALCRL identification in this study. Accordingly, we have clarified and simplified this point throughout the revised manuscript for the broad audience readers of *Nature Communications*.

Finally, while CALCRL is a novel prognostic marker in AML, its biological function in AML (and other cancers indeed) is still largely unknown. Here, our study confirmed the clinical aspect and interest of this cell surface target in two independent clinical cohorts with two different techniques, and deeply addressed the role of CALCRL axis in the function of RICs in AML *in vitro*, *in vivo* and in patients. We are convinced that this manuscript will markedly bring out key aspects of CALCRL in AML/cancer biology, discovery and therapies as promising target.

Reviewer #3 (Remarks to the Author):

The response to the critiques is satisfactory and the paper is improved. one point remains- for western blots, the protein loading control should be done on the same gel. The authors are requested to repeat the immunoblot and do the controls on the same gel.

We thank the reviewer. We incorporated uncropped Western-Blot in the end of the supplementary figure document. Moreover, we certified that the loading control was carried out on a membrane which has been previously used to reveal another protein (except for the two western-blots carried out on primary samples):

- Figure 4d: Actin was from BCL2 membrane (MOLM14) or RAD51 membrane (OCI-AML3)
- Figure 4i: Actinin was added to cyclin A membrane.
- Figure 4j: Actin was added to CHK1 membrane (MOLM14 and OCI-AML3)
- Figure 4q: For Patients 6 and 19, to reveal Actin, we used samples deposited in parallel on the same time.
- Figure 5f: Actin was added to BCL2 membrane